# Diversity of the Origin of Cancer Stem Cells in Oral Squamous Cell Carcinoma and Its Clinical Implications

**DOI:** 10.3390/cancers14153588

**Published:** 2022-07-22

**Authors:** Chonji Fukumoto, Daisuke Uchida, Hitoshi Kawamata

**Affiliations:** 1Department of Oral and Maxillofacial Surgery, Dokkyo Medical University School of Medicine, 880 Kitakobayashi, Shimo-Tsuga, Mibu 321-0293, Tochigi, Japan; chonji-f@dokkyomed.ac.jp; 2Department of Oral and Maxillofacial Surgery, Ehime University Graduate School of Medicine, 454 Shitsukawa, Toon 791-0295, Ehime, Japan; udai@m.ehime-u.ac.jp

**Keywords:** oral squamous cell carcinoma, origin of cancer, stem cell, cancer stem cell, somatic stem cell, oral epithelial stem cell, bone marrow-derived stem cell

## Abstract

**Simple Summary:**

Oral squamous cell carcinoma (OSCC) histopathologically accounts for ≥90% of oral cancer. In this review article, we focus on the diversity of the origin of OSCC and also discuss cancer stem cells (CSCs). CSCs are a subset of cancer cells that occupy a very small portion of the cancer mass and have characteristics of stem cells. When gene abnormalities accumulate in somatic stem cells, those cells transform into CSCs. CSCs as the origin of cancer then autonomously grow and develop into cancer. The histopathological phenotype of cancer cells is determined by the original characteristics of the somatic stem cells and/or surrounding environment. OSCC may be divided into the following three categories with different malignancy based on the origin of CSCs: cancer from oral epithelial stem cell-derived CSCs, cancer from stem cells in salivary gland-derived CSCs, and cancer from bone marrow-derived stem cell-derived CSCs.

**Abstract:**

Oral squamous cell carcinoma (OSCC) histopathologically accounts for ≥90% of oral cancer. Many clinicopathological risk factors for OSCC have also been proposed, and postoperative therapy is recommended in guidelines based on cancer stage and other risk factors. However, even if the standard treatment is provided according to the guidelines, a few cases rapidly recur or show cervical and distant metastasis. In this review article, we focus on the diversity of the origin of OSCC. We also discuss cancer stem cells (CSCs) as a key player to explain the malignancy of OSCC. CSCs are a subset of cancer cells that occupy a very small portion of the cancer mass and have characteristics of stem cells. When gene abnormalities accumulate in somatic stem cells, those cells transform into CSCs. CSCs as the origin of cancer then autonomously grow and develop into cancer. The histopathological phenotype of cancer cells is determined by the original characteristics of the somatic stem cells and/or surrounding environment. OSCC may be divided into the following three categories with different malignancy based on the origin of CSCs: cancer from oral epithelial stem cell-derived CSCs, cancer from stem cells in salivary gland-derived CSCs, and cancer from bone marrow-derived stem cell-derived CSCs.

## 1. Introduction

Oral cancer is a general term for malignant tumors developing in the oral cavity and is the eighteenth most common cancer [1,2]. Every year, there are 377,713 cases of newly diagnosed oral cancer and 177,757 deaths from this cancer [2,3]. According to the definition from the Union for International Cancer Control, oral cancer includes cancers developing on the buccal mucosa, maxillary gingiva, mandibular gingiva, hard palate, tongue, oral floor, and lips [4]. Since the oral mucosa is covered by stratified squamous epithelium, squamous cell carcinoma (SCC) histopathologically accounts for ≥90% of cases of oral cancer [5]. Other types of oral cancer include salivary gland cancer, malignant melanoma, malignant lymphoma, sarcoma, and metastatic cancer. Oral squamous cell carcinoma (OSCC) is mainly treated surgically and also with a combination of radiation and chemotherapeutic drugs. In recent years, the usage of molecular-targeted drugs and immune checkpoint inhibitors has increased [6,7,8,9,10]. Rapid progress with these therapeutic methods has improved the outcomes for OSCC, but local recurrence, lymph node metastasis, and distal metastasis still occur at a certain rate. Local recurrence may occur even after complete surgical resection with an appropriate safety margin. In addition, in cases that are evaluated as negative for lymph node metastasis by preoperative examinations with highly accurate modalities such as fluorodeoxyglucose-positron emission tomography (FDG-PET), contrast computed tomography (CT), magnetic resonance imaging (MRI), and ultrasonography, cervical lymph node metastasis may subsequently develop, which decreases the survival rate [7,8].

It is well-known that the prognosis of oral cancer correlates with cancer stage [11], and many other clinicopathological risk factors have also been proposed [12,13,14,15,16,17,18,19,20,21,22,23,24,25]. Postoperative radiation therapy and chemotherapy are recommended based on cancer stage and other risk factors in several guidelines [12]. However, even if the standard treatment is provided according to the guidelines, tumor recurrence and metastasis rapidly occur in a few cases. For example, there is a case of early-stage oral cancer with a minute primary tumor without any detectable metastasis that shows a poor prognosis after complete resection of the primary tumor, due to cervical lymph node metastasis and distal metastasis in an extremely early phase (Figure 1: A 60-year-old male patient with right tongue SCC. Tumor diameter was 7 × 5 mm, depth of invasion was 1 mm (T1N0M0), and degree of differentiation was moderate. Surgical margins were negative.). On the other hand, there is a case of advanced oral cancer with significant local proliferation and invasion that shows a favorable prognosis after appropriate resection of the primary tumor (Figure 2: A 65-year-old male patient with right tongue SCC. Tumor diameter was 62 × 39 mm, depth of invasion was 30 mm, bilateral cervical lymph node metastasis (T4aN2cM0), and degree of differentiation was moderate. Surgical margins were negative.). This raises the questions of, “Two such cases are the same disease”?

Cancer stage is determined at the first hospital visit, and the diagnosis is made without consideration of factors related to time course until the time of diagnosis. At the first visit, cancer stage may be determined based on a product of “biological malignancy of the tumor” and “time since its occurrence”. This means that if the time from occurrence is long at the first hospital visit, even a tumor with low biological malignancy (proliferative, invasive, and metastatic capacities) may be diagnosed as an advanced large tumor. On the other hand, a tumor with extremely high biological malignancy that is diagnosed immediately after its occurrence may be defined as a cancer at an early stage. Because the primary lesion of such a tumor is small, the invasion to surrounding tissues is mild, and lymph node or distal metastasis is too small to be detected by imaging. To resolve such problems, an approach to determining fundamental malignancy using a genetic analysis of the tumor cells has been suggested [26,27,28,29,30], but no target molecules and standard methods have been established at this point.

We have focused on the diversity of the origin of OSCC and on cancer stem cells (CSCs) [31,32,33,34,35,36,37,38,39,40] as a key plyer to explain the malignancy of OSCC. CSCs are a subset of cancer cells that occupy a very small portion of the tumor mass and have characteristics of stem cells [34,35,37,40,41]. CSCs have a capability of asymmetrical cell division, and the self-renewal capacity of CSCs produces a heterogeneous population of cancer cells. Furthermore, CSCs show higher anti-apoptosis activity, invasiveness, and survivability in the ectopic environment than other subsets of cancer cells. Thus, CSCs have more malignant characteristics, including tumorigenicity and metastatic potential, than other subsets of cancer cells. In general, when gene abnormalities accumulate in somatic stem cells, those cells transform into CSCs. CSCs then autonomously grow and develop into cancer [31,33,42,43,44]. The histopathological phenotype of cancer cells is determined by the original characteristics of the somatic stem cells and/or surrounding environment; then, some cells become squamous cancer, adenocarcinoma, sarcoma, lymphoma, or leukemia. When gene abnormalities accumulate in more undifferentiated and multipotent somatic stem cells, such as bone marrow stem cells, transformed somatic stem cells (CSCs) can form tumors with several histopathological phenotypes depending on the surrounding environments.

In this review article, we describe the general features of CSCs and discuss the diversity of origin of the CSCs in OSCC based on our published results. We also discuss the evaluation of biological malignancy of OSCC based on the origin of CSCs and the selection of an appropriate treatment strategy for each OSCC.

## 2. CSCs in OSCC

### 2.1. Cancer Stem Cells (CSCs)

Previous studies have suggested that cancers are organized as a cellular hierarchy of various populations of cancer cells, and the hierarchy is maintained by a small subset of cancer cells, called CSCs [33,34]. CSCs have stemness and are also called cancer-initiating cells [31,32,33,34,35,36,37,38,39,40,41,42,45]. CSCs were identified for the first time in 1997 by Bonnet and Dick in a patient with acute myeloid leukemia [46]. In a tumor transplant experiment in immunodeficient mice, only a small subset of leukemia cells with CD34^+^/CD38^−^ could regenerate the original leukemia [46]. Then, in 2003, Al-Hajj et al. reported that only a subset of CD44^+^/CD24^−^ breast cancer cells could reestablish a tumor mass with the same phenotype from the original tumor, leading to the identification of CSCs in solid cancer [47]. Subsequently, CSCs have been found in many types of solid cancer [33,42]. CSCs are characterized by abilities such as long-term potential for self-renewal, proliferation, and differentiation in a tumor mass [34,48,49]. In addition, CSCs have many biological characteristics, including resistance to chemotherapy and radiation therapy [50,51], avoidance of induced cell death [52,53], and dormancy [54], which differ from those of non-CSCs. Based on these characteristics, CSCs are thought to play a leading role in the development, progression, and recurrence of cancer.

### 2.2. CSCs in OSCC

CSCs have also been shown to play important roles in the development and progression of OSCC [31,32,33,34,35,36,37,38,39,40]. It is reported that only a small percentage of the cell population contributes to the development of OSCC, and this population has properties corresponding to those of CSCs [31,40]. These findings suggest that the OSCC tumor mass is a mixture of: (1) CSCs with the capabilities of asymmetrical cell division and self-renewal, (2) transient proliferating cells (progenitor cells) with the capability of several self-divisions, and (3) differentiated cells that do not contribute to tumor proliferation [31]. There are no specific single markers that clearly define CSCs in OSCC, but several markers are combined to identify CSCs in OSCC. Some CSCs express similar proteins to those that regulate embryonic stem cells (ESCs), including OCT4, NANOG, and SOX2, which are used as markers for CSCs in OSCC [45]. These proteins are master regulators of self-renewal and maintenance in the undifferentiated stem cell population. CD44, CD24, CD133, Musashi-1, and ALDH have also been reported as potential markers of CSCs in OSCC [31,35,38,39,40].

## 3. Possible Candidates of the Origin of CSCs in OSCC

### 3.1. Oral Epithelial Stem Cells Located Close to Basal Cells of Stratified Squamous Epithelium

Oral epithelial stem cells, which are located close to the basal cells of stratified squamous epithelium, can be a candidate for the origin of CSCs in OSCC [31,33,36]. Oral epithelium is formed with several cell layers, including the basal, prickle cell, granular, and cornified layers in the keratinizing region, and the basal, prickle cell, inner, and surface layers in the non-keratinizing region. Cell proliferation occurs in the basal cell layer or supra-basal layer to supply new cells; the cells differentiate toward the surface layer, and the components of the entire epithelial mucosa are renewed in 5 to 40 days. This high regenerative ability and rapid replacement of tissues in the oral mucosa are related to the presence and dynamics of oral epithelial stem cells in the basal cell layer. Oral epithelial stem cells expressing Bmi1 were first reported in the dorsal tongue epithelium [55,56]. Transformation from oral epithelial stem cells to CSCs and OSCC development may involve the activation of K-ras, a combination of activated K-ras and the loss of p53, the loss of SMAD4, the double loss of PTEN and TGF-β receptor type 1, and the deregulation of NOTCH signaling [33,36,38,39,41].

### 3.2. Stem Cells in Minor Salivary Gland That Can Differentiate to Salivary Gland Components, Squamous Cells, or Mesenchymal Cells

Our research group proposed the intriguing hypothesis that OSCC may be composed of at least two different cancers: mucosal-SCC derived from squamous epithelium and salivary-SCC derived from the minor salivary gland [44]. These two origins of OSCC were defined using a clustering analysis of whole gene expression profiles, and nine genes were identified as markers of salivary-SCC. Furthermore, we revealed that salivary-SCC showed high metastatic potential and a poor event-free survival rate.

TYS cells, a cell line established using well-differentiated SCC of the oral floor, showed high expression of carcinoembryonic antigen, a tumor marker for adenocarcinoma, and formed adenosquamous cancer in nude mice [57]. HSG cells established from an irradiated submandibular gland [58] also form adenocarcinoma in nude mice. HSG cells differentiate into several different cell types (myoepithelial cells, acinic cells, chondrocyte-like cells, bone-forming cells, neuronal-like cells) upon treatment with various differentiation-inducing agents [59,60,61,62,63,64,65]. HSG cells can also differentiate into keratinizing SCC after treating with retinoic acid [66]. These reports also provide experimental support for the possibility that some salivary gland cells may be the origin of SCC in the oral cavity. Salivary gland cancer is generally recognized to be more resistant to radiation and chemotherapy compared to SCC and to metastasize to distal organs, causing a poor prognosis [67,68]. Therefore, for salivary-SCC in the oral cavity, a wider resection margin than that for mucosal-SCC in the oral cavity and concomitant use of aggressive radical neck dissection for suspicious cervical lymph nodes may be desirable.

### 3.3. Bone Marrow-Derived Pluripotential Stem Cells in Peripheral Blood

It has been reported that some patients develop OSCC from oral lichenoid lesions caused by chronic graft-versus-host disease after hematopoietic stem cell transplantation in the treatment of hematopoietic malignancies [69,70]. As a hypothesis to explain these cases, Hasegawa et al. in our research group proposed that bone marrow (BM)-derived stem cells could be the origin that causes the development of OSCC [43]. Sex-chromosome analysis with fluorescence in situ hybridization and microdissection PCR of samples from patients who developed OSCC after hematopoietic stem cell transplantation from a person of the opposite sex suggested that OSCC was caused by donor-derived BM cells and showed poor prognosis [43]. In microarray and cluster analyses of OSCC cases with possible BM stem cell-derived OSCC, several candidate marker genes were identified [43]. Recent research on the development of regenerative medicine has identified many somatic stem cells. For example, the stem cells in hepatocyte and vascular endothelial cells were demonstrated to be of bone marrow (BM) origin and were present in peripheral blood [71,72]. It was also suggested that BM-derived stem cells may be involved in the regeneration of skin and gastrointestinal epithelial cells [73,74].

It is well known that BM stem cells include hematopoietic progenitors and mesenchymal stem cells, and that multilineage-differentiating stress-enduring (Muse) cells, a small group of BM mesenchymal stem cells in circulating blood, can differentiate into several types of cells, including epithelial cells [75]. In contrast to ES cells and iPS cells, Muse cells are pluripotent cells with no possibility of neoplastic transformation, and thus, BM-derived stem cells related to the development of OSCC may be a mass of non-Muse mesenchymal stem cells. Kano et al. reported similar results for patients with esophageal SCC developed from BM-derived stem cells [76], and BM-derived stem cells are one of the leading candidates as an origin in the development of SCCs.

## 4. Multistep Carcinogenesis, Fundamentally Good or Fundamentally Evil?

A mechanism in which genetic abnormalities accumulate in somatic stem cells resulting in carcinogenesis (multistep carcinogenesis) has been proposed [77,78]. We have suggested that somatic stem cells with genetic abnormalities (these are CSCs) could be the origin of OSCC and may be composed of oral epithelial stem cell-derived CSCs, stem cells in salivary gland-derived CSCs, and BM-derived stem cell-derived CSCs (Figure 3). Recently, in a systematic review and meta-analysis, de la Cour et al. reported that 27% of oral epithelial dysplasia had Human papillomavirus (HPV)-DNA [79]. However, the detailed role of HPV in carcinogenesis from oral epithelial dysplasia to OSCC remained unclear. We previously reported that the constitutive expressions of HPV16 E6/E7 as a result of HPV integration in the genome was observed in 15.5% of OSCC, while there was no correlation with prognosis and p53 abnormalities [80]. There were also negative results regarding the involvement of HPV in OSCC carcinogenesis [81]. At present, we consider that oncogenic HPV infection and integration may be associated with the carcinogenesis of CSCs from oral epithelial stem cells. Further research and discussion may be needed to determine the association of HPV in the carcinogenesis of OSCC, especially in our three different, hypothetical origins of CSCs.

Cancer that develops from more undifferentiated stem cell-derived CSCs is considered to shows “stemness” and high malignancy (invasion, metastatic potential). BM-derived stem cell-derived CSCs are more undifferentiated than oral epithelial stem cell-derived CSCs or stem cells in salivary gland-derived CSCs. Conventionally, most cases of OSCC have been viewed as developing through “gain of function” as an accumulation of genetic abnormalities in oral epithelial stem cells or salivary gland stem cells, gaining the potential for proliferation, invasion, and metastasis [77]. This idea may be proposed based on the concept of “fundamentally good”, in which normal cells do not behave similarly to malignant cells (that is, infinite proliferative capacity, tissue infiltration by epithelial–mesenchymal transition, evasion from drugs by hiding in a niche, efflux of drugs from inside of cells, anti-apoptosis, anti-terminal differentiation, angiogenic potential, survival in blood vessels, and survival and proliferation in an ectopic environment). In contrast, we would like to propose that the development of highly malignant cancer from more undifferentiated stem cell-derived CSCs is based on the concept of “fundamentally evil.” The above characteristics of malignant cells are consistent with the stemness of pluripotent stem cells, such as BM-derived stem cells. Pluripotent stem cells with these characteristics behave as differentiated normal cells in their environment, masking their stemness, probably by the methylation of the genes. When genetic abnormalities accumulate in pluripotent stem cells and they become CSCs, their original masked characteristics, i.e., stemness, are exhibited, and the tumor cells show extremely malignant behavior. This can be explained by the hypothesis that there is another multistep carcinogenesis model that allows the malignant transformation to proceed by gain of oncogenic function and the loss of masked stemness, i.e., “loss of function” (Figure 4).

To summarize our hypothesis, we propose that there are at least three CSCs that could be considered as the origin in OSCC and that the characteristics of CSCs can be divided into “fundamentally good” or “fundamentally evil”. Among the three CSCs, oral epithelial stem cell-derived CSCs are classified as “fundamentally good”, BM-derived stem cell-derived CSCs as “fundamentally evil”, and stem cells in salivary gland-derived CSCs as intermediate between the two, based on their stemness. If we can identify the type of CSCs as the origin of OSCC at the time of pre-treatment staging, the true malignancy of each individual tumor may be determined. If some tumors can be diagnosed as “fundamentally evil” by our new concept without being guided by the conventional staging of the tumor, it would be desirable to apply escalated surgical resection and intensive postoperative treatment. Alternatively, if some tumors can be diagnosed as “fundamentally good”, the excessive surgical resection and aggressive postoperative therapy may be avoided. This diagnosis concept based on the origin of CSCs for OSCC leads to a paradigm shift from a conventional diagnosis system and treatment strategy based on clinical stage and histopathology alone. Further research may be necessary to develop high-throughput and more accurate methods to identify the type of CSCs as the origin of individual OSCC.

## 5. Conclusions

OSCC may be divided into the following three categories based on the origin of CSCs: cancer form oral epithelial stem cell-derived CSCs, cancer from stem cells in salivary gland-derived CSCs, and cancer from BM-derived stem cell-derived CSCs. Diagnosis of OSCC based on the origin of CSCs is likely to be helpful for the development of concrete therapeutic strategies to improve prognosis through the prediction of potent malignancy. This idea may represent a paradigm shift in diagnostic and therapeutic strategies for OSCC.

## Figures and Tables

**Figure 1 cancers-14-03588-f001:**
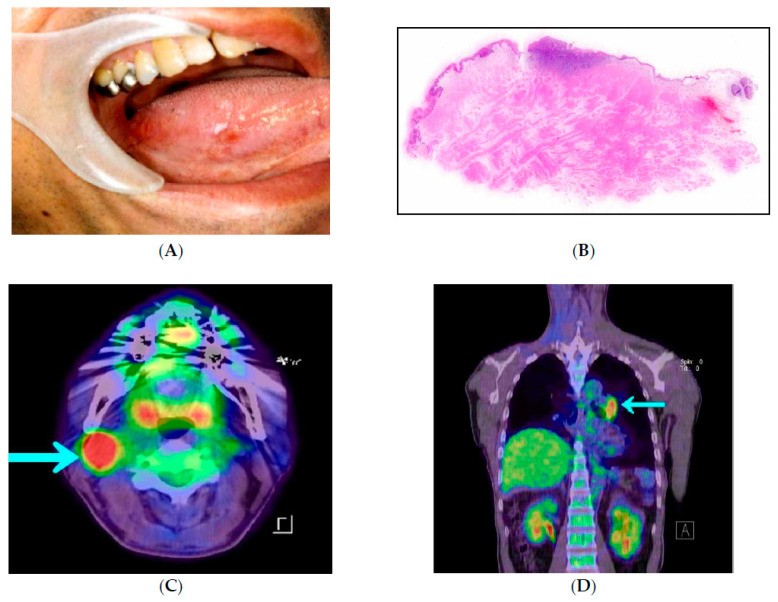
A 60-year-old male patient underwent partial glossectomy for right SCC of the tongue (T1N0M0, tumor diameter was 7 × 5 mm, depth of invasion was 1 mm, and degree of differentiation was moderate; surgical margins were negative). Cervical metastasis was confirmed 12 months after the initial surgery, and radical neck dissection was performed. Pulmonary metastasis occurred 3 months later, and the patient died due to the disease. (**A**) Photograph of the oral cavity at the first hospital visit. (**B**) Magnified image of a sample from partial glossectomy. (**C**) FDG-PET/CT at 12 months after the initial surgery. (**D**) FDG-PET/CT at 3 months after radical neck dissection.

**Figure 2 cancers-14-03588-f002:**
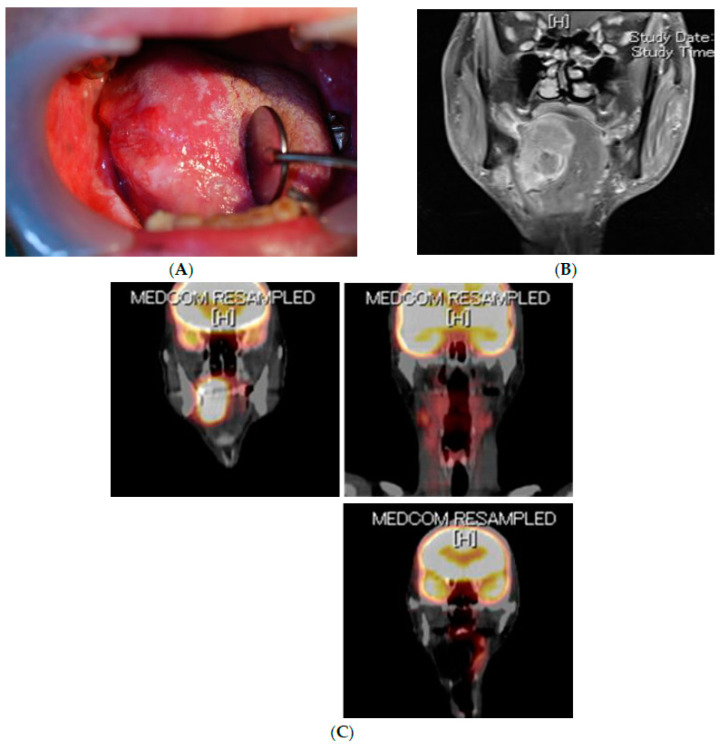
A 65-year-old male patient with right SCC of the tongue (T4aN2cM0, tumor diameter was 62 × 39 mm, depth of invasion was 30 mm, and degree of differentiation was moderate; surgical margins were negative) underwent bilateral radical neck dissection, total removal of the tongue, and reconstruction with a rectus abdominis musculocutaneous flap. The patient has survived and been disease-free for ≥5 years since the surgery. (**A**) Photograph of the oral cavity at the first hospital visit. (**B**) MRI at the first hospital visit. (**C**) FDG-PET/CT at the first hospital visit.

**Figure 3 cancers-14-03588-f003:**
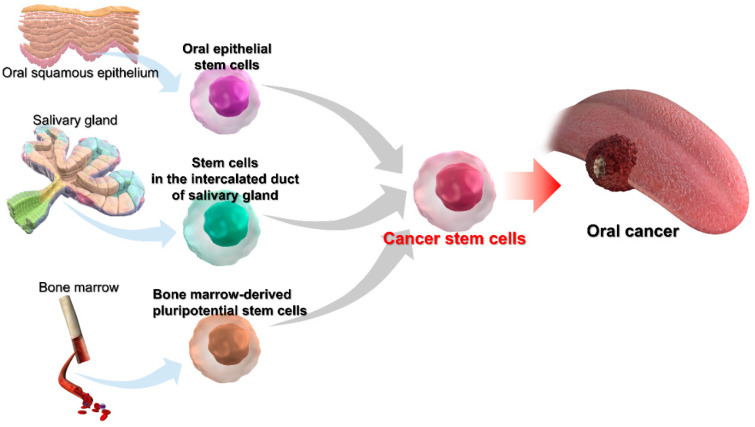
The illustration of the development of OSCC by several CSCs.

**Figure 4 cancers-14-03588-f004:**
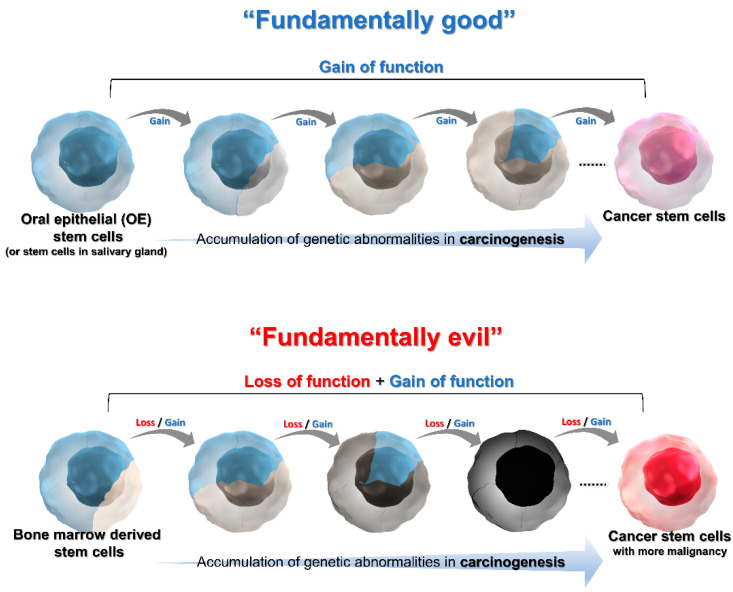
The illustration of “Fundamentally good” as conventional multistep carcinogenesis model and “Fundamentally evil” as newly proposed multistep carcinogenesis model.

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
