# Peer review of "Diversity of the Origin of Cancer Stem Cells in Oral Squamous Cell Carcinoma and Its Clinical Implications"

_cancers, 2022, doi:10.3390/cancers14153588_

Round 1
Reviewer 1 Report
This article reviewed the cancer stem cells of oral squamous cell carcinoma from the different origins. Based on the developmental region of the cancer, cancer stem cells (CPCs) were classified into epithelial stem cells-derived CSCs, salivary gland-derived CSCs and bone marrow-derived stem cells-derived CSC, and the characteristics of each cell were described in this article. The important thing is what kind of property the tumor has when malignant tumor is treated. The tissue type of the tumor is one of the key factors to evaluate the prognosis. However, even in the same tissue type, there are various types such as those that easily grow inward, those that easily metastasize to regional lymph nodes, and those that easily cause distant metastasis. It is very difficult to decide the character of the malignant tumor. Clarifying how CSC is related to the properties of the tumor, prognosis after treatment, and therapeutic effect on the tumor will make this paper more impactful. In addition to squamous cell carcinoma, various cancers develop in oral cancer. In particular, since salivary gland primary has more adenocarcinoma than squamous cell carcinoma, consideration to adenocarcinoma is necessary.
Author Response
Response to the Reviewer 1.
Thank you very much for your warm comments. Our responses are as follows:
- Q) Clarifying how CSC is related to the properties of the tumor, prognosis after treatment,
Response) According to the review’s comments, we added new descriptions concerning the biological malignancy, clinical behavior, and prognosis of each OSCC from different origin. Furthermore, we mentioned that this diagnosis system (or concept) based on the origin of CSC for OSCC lead a paradigm shift from a conventional diagnosis system and treatment strategy based on clinical stage and histopathology alone. (Page 7, line 52)
Response) In addition to squamous cell carcinoma, various cancers develop in oral cancer. In particular, since salivary gland primary has more adenocarcinoma than squamous cell carcinoma, consideration to adenocarcinoma is necessary.
- Q) In addition to squamous cell carcinoma, various cancers develop in oral cancer. In particular, since salivary gland primary has more adenocarcinoma than squamous cell carcinoma, consideration to adenocarcinoma is necessary.
Response) In this manuscript, we focused on SCCs in the oral cavity. We did not discuss the primary malignancies including adeno-type carcinoma of the major salivary glands. If we identified malignant tumors with adeno-like phenotype in oral cavity, we treated such tumors as minor salivary gland tumors. If we have a chance, we will write another review article on the characteristics of various histologic types of salivary gland tumor.
Reviewer 2 Report
The paper provides an interesting hypothesis, which is quite well supported by previous research. The importance of the paper is underscored by some inconsistencies presented by the WHO 2022 Tumor classification - oral dysplasia and oral cancer chapters. Further research into oral cancer dysplasia and carcinogenesis is needed. This paper shows a new direction. At the same time, maybe it could allow some space for anticipated future developments, especially in regard to possible HPV-associated carcinogenesis. In this way, the paper may remain relevant in the time to come.
In my opinion, the work is very original, truly visionary, aiming at the "knowledge gap area" and worth publishing.

Author Response
Response to the Reviewer 2.
Thank you very much for your warm comments. Our responses are as follows:
【Page 1】
- Q) the misspelling of "from"
Response) We corrected.
【Page 2】
- Q) Please, specify this citation properly. Preferably, cite and apply WHO 2022 if possible. WHO Classification of Tumours Editorial Board. Head and neck tumours [Internet; beta version ahead of print]. Lyon (France): International Agency for Research on Cancer; 2022 [cited YYYY Mmm D]. (WHO classification of tumours series, 5th ed.; vol. 9). Available from: https://tumourclassification.iarc.who.int/chapters/52.
Response) In response to your suggestion, we cited the WHO Blue Book 2022 (web version). We also added it to the references. (Ref, No. 3)
- Q) Please, also in the main manuscript (not only as a figure legend) specify the size, depth of invasion, completness of surgical excision.
- Q) Please supply the stage as per UICC 2018.
- Q) Please specify the grade and tumor histomorphology, depth of invasion, presence of angioinvasion, surgical margins status, supported by proper stage designation as per UICC 2018. Also, it is important to compare both cases in regard of p16 expression and HPV status which is known to influence the prognosis of squamous cell carcinoma in Head and Neck area.
- Q) Here, is the place which need Comment on differences caused by possible HPV associated pathogenesis.
Response) Yes, we added the clinical information for each case in the text and figure legends. Unfortunately, we did not examine the expression of p16 as well as the constitutive expressions of E6 and E7 in these two cases. We reported the oncogenic HPV integration and constitutive expression of E6 and E7 in carcinogenesis of OSCC before this experiment. According to the reviewer’s suggestion, we added the discussion of association of HPV infection on OSCC carcinogenesis based on the others’ paper and our previous paper. (Page 7, Line 18)
【Page 5】
- Q) This interesting suggestion should be recalibrated based on the possibility of HPV associated cancarogenesis. So, maybe it would be apropriate to point to the fact that, probably, this hypothesis was not tested for HPV factor.
【Page 6】
- Q) This hypothesis should be adjusted to the notion, that some oral cancers are preceded by HPV associted dysplasia. Though, its exact role and time-sequence is unknown.
【Page 7】
- Q) Please, make sure your hypothesis reflects (allows some space) for possible future developement regarding oral squamous carcinoma arrisig from HPV associated oral dysplasia representing up to 27% of oral dysplasia cases.
de la Cour CD, Sperling CD, Belmonte F, Syrjänen S, Verdoodt F, Kjaer SK. Prevalence of human papillomavirus in oral epithelial dysplasia: Systematic review and meta-analysis. Head Neck. 2020 Oct;42(10):2975-2984. doi: 10.1002/hed.26330. Epub 2020 Jun 23. PMID: 32573035.
【Page 8】
- Q) Please, make sure your hypothesis allows some space for possible future developement regarding oral squamous carcinoma arrisig from HPV associated oral dysplasia representing up to 27% of oral dysplasia cases.
de la Cour CD, Sperling CD, Belmonte F, Syrjänen S, Verdoodt F, Kjaer SK. Prevalence of human papillomavirus in oral epithelial dysplasia: Systematic review and meta-analysis. Head Neck. 2020 Oct;42(10):2975-2984. doi: 10.1002/hed.26330. Epub 2020 Jun 23. PMID: 32573035.
Response) Thank you for useful suggestions. We cited the suggested systematic review to the references (Ref. No. 79). We also added our previous paper concerning HPV in OSCC. Based on these papers, we added the description of HPV in the section of "4) Multistep carcinogenesis, fundamentally good or fundamentally evil?" as follows:
"Recently, in the systematic review and meta-analysis, de la Cour et al. reported that 27% of oral epithelial dysplasia had Human papillomavirus (HPV)-DNA [79]. However, the detailed role of HPV in carcinogenesis from oral epithelial dysplasia to OSCC remained unclear. We previously reported that the constitutive expressions of HPV16 E6/E7 as a result of HPV integration in the genome was observed in 15.5% of OSCC, while there was no correlation with prognosis and p53 abnormalities [80]. There were also negative results on the involvement of HPV in OSCC carcinogenesis [81]. At present, we considered that oncogenic HPV infection and integration might be associated with carcinogenesis on CSCs from oral epithelial stem cells. Further research and discussion may be needed to determine the association of HPV in carcinogenesis of OSCC, especially in our hypothetical three different origin of CSCs.”
Reviewer 3 Report
The article presentation is appropriate, although the authors define the study as a review article, it is not and this should be fixed in the text. The presentation of two clinical examples of OSCC should not be in fact in the introduction of a review article as indicated, especially without references. This is a commentary article and not a review (in fact all the other parts of a classical paper are missing). The title of the commentary should be more focused on the stem-cell origin of oral cancer.
As aims of the paper, the Authors declare "In this review article, we describe the general features of CSCs and discuss the diversity of origin of the CSCs in OSCC, based on our published results". The rationale of this assumption is not clear, and, in any case, other studies of the literature also seem to have been considered. It would be useful to clarify this aspect.
Moreover, with regard to the concepts of “fundamentally good” and “fundamentally evil”, the theory should be further deepened, better explaining what is meant by the gain and loss of functions in relation to different types of stem cells, making the text easier to understand at first reading. Authors in this regard should pay also attention to all the acronyms in the text: each should always be accompanied by writings in full for greater understanding, inserted where absent. The division into sections is accurate and optimal (no changes required) and the images are very functional to the text, but the text on carcinogenesis theory, as mentioned above, should be implemented and more characterized. In this way of view, the conclusion should be more extensive in relation to the consequences that such theory could have on the diagnosis and treatment of oral cancer, explaining why - as cited by them - "This idea may represent a paradigm shift in diagnostic and therapeutic strategies for OSCC".
Author Response
Response to the Reviewer 3.
Thank you very much for your warm comments. Our responses are as follows:
Q) The article presentation is appropriate, although the authors define the study as a review article, it is not and this should be fixed in the text. The presentation of two clinical examples of OSCC should not be in fact in the introduction of a review article as indicated, especially without references. This is a commentary article and not a review (in fact all the other parts of a classical paper are missing). The title of the commentary should be more focused on the stem-cell origin of oral cancer.
Response)Yes, we fully understand the reviewer's comments. We are originally invited to write this article as a “narrative review”, in which we describe a general concept of cancer stem cells on OSCC and a new hypothesis (mostly proven facts by our group) about the diversity of origin of cancer stem cells for OSCC. This article is now classified as a “Commentary” according to the word count rule of this journal. Unlike Systematic review, there might be no typical style for narrative review or commentary. Therefore, we first show two high-impact cases with quite different biological characteristics and clinical outcome to strongly demonstrate the diversity of OSCC. These two cases might support our logic and hypothesis. Next, we described the clinical and molecular epidemiology of the existence of at least three types of cancer stem cells as the origin of OSCC. Finally, we discussed the hypotheses of fundamentally good and fundamentally evil based on these results. According to the reviewer’s suggestion, the title was changed to "Diversity of the origin of cancer stem cells in OSCC and its clinical implications".
Q) As aims of the paper, the Authors declare "In this review article, we describe the general features of CSCs and discuss the diversity of origin of the CSCs in OSCC, based on our published results". The rationale of this assumption is not clear, and, in any case, other studies of the literature also seem to have been considered. It would be useful to clarify this aspect.
Response) The general features of somatic stem cells and CSCs were described in the previously published papers. However, the diversity of origin of the CSCs in OSCC has not been reported by any other research groups. In other organs, some studies have been reported on the diversity of origin cell in tissue regeneration and carcinogenesis. We cited these papers and added new description. (Page 6, line 49, and Ref. No. 71-74)
Q) Moreover, with regard to the concepts of “fundamentally good” and “fundamentally evil”, the theory should be further deepened, better explaining what is meant by the gain and loss of functions in relation to different types of stem cells, making the text easier to understand at first reading. Authors in this regard should pay also attention to all the acronyms in the text: each should always be accompanied by writings in full for greater understanding, inserted where absent.
Response) We have tried to explain the concept of “fundamentally good” and “fundamentally evil” clearly enough in the original manuscript. However, after hearing the reviewer’s comments, we modified the description to make it even clearer and more understandable(Page 7, line 52). We also changed the abbreviations to their full spelling except OSCC, CSC, and BM.
Q) The division into sections is accurate and optimal (no changes required) and the images are very functional to the text, but the text on carcinogenesis theory, as mentioned above, should be implemented and more characterized. In this way of view, the conclusion should be more extensive in relation to the consequences that such theory could have on the diagnosis and treatment of oral cancer, explaining why - as cited by them - "This idea may represent a paradigm shift in diagnostic and therapeutic strategies for OSCC".
Response) According to the review’s comments, we added new descriptions concerning the biological malignancy, clinical behavior, and prognosis of each OSCC from different origin. Furthermore, we mentioned that this diagnosis system (or concept) based on the origin of CSC for OSCC lead a paradigm shift from a conventional diagnosis system and treatment strategy based on clinical stage and histopathology alone. (Page 7, line 52)
Round 2
Reviewer 1 Report
Now, the text was completely reconstructed and reached the level for publication in Cancers.
Reviewer 3 Report
The revised version has been sufficiently improved.